# The Minimax Rate of HSIC Estimation for Translation-Invariant Kernels

**Florian Kalinke**
Institute for Program Structures and Data Organization
Karlsruhe Institute of Technology
Karlsruhe, Germany
florian.kalinke@kit.edu

**Zoltán Szabó**
Department of Statistics
London School of Economics
London, UK
z.szabo@lse.ac.uk

## Abstract

Kernel techniques are among the most influential approaches in data science and statistics. Under mild conditions, the reproducing kernel Hilbert space associated to a kernel is capable of encoding the independence of $M \geq 2$ random variables. Probably the most widespread independence measure relying on kernels is the so-called Hilbert-Schmidt independence criterion (HSIC; also referred to as distance covariance in the statistics literature). Despite various existing HSIC estimators designed since its introduction close to two decades ago, the fundamental question of the rate at which HSIC can be estimated is still open. In this work, we prove that the minimax optimal rate of HSIC estimation on $\mathbb{R}^d$ for Borel measures containing the Gaussians with continuous bounded translation-invariant characteristic kernels is $\mathcal{O}(n^{-1/2})$. Specifically, our result implies the optimality in the minimax sense of many of the most-frequently used estimators (including the U-statistic, the V-statistic, and the Nyström-based one) on $\mathbb{R}^d$.

## 1 Introduction

Kernel methods [Steinwart and Christmann, 2008, Berlinet and Thomas-Agnan, 2004, Saitoh and Sawano, 2016] allow embedding probability measures into reproducing kernel Hilbert spaces (RKHS; [Aronszajn, 1950]) by use of a positive definite function, the *kernel function*. This approach has gained considerable attention over the last 20 years. Such embeddings induce the so-called maximum mean discrepancy (MMD; [Smola et al., 2007, Gretton et al., 2012]), which quantifies the discrepancy of two probability measures by considering the RKHS norm of the distance of their respective embeddings. MMD is a metric on the space of probability distributions if the kernel is characteristic [Fukumizu et al., 2008, Sriperumbudur et al., 2010]. MMD is also an integral probability metric [Zolotarev, 1983, Müller, 1997] where the underlying function class is chosen to be the unit ball in the corresponding RKHS.

MMD allows for the quantification of dependence by considering the distance between the embedding of a joint distribution and that of the product of its marginals. This construction gives rise to the so-called Hilbert-Schmidt independence criterion (HSIC; [Gretton et al., 2005]), which is also equal to the RKHS norm of the centered cross-covariance operator. In fact, one of the most widely-used independence measures in statistics, distance covariance [Székely et al., 2007, Székely and Rizzo, 2009, Lyons, 2013], was shown to be equivalent to HSIC [Sejdinovic et al., 2013b] when the latter is specialized to $M = 2$ components; Sheng and Sriperumbudur [2023] proved a similar result for the conditional case. For $M > 2$ components [Quadrianto et al., 2009, Sejdinovic et al., 2013a, Pfister et al., 2018], universality [Steinwart, 2001, Micchelli et al., 2006, Carmeli et al., 2010, Sriperumbudur et al., 2011] of the kernels $(k_m)_{m=1}^M$ (on the respective domains) underlying HSIC guarantees that this measure captures independence [Szabó and Sriperumbudur, 2018]. In the case of $M = 2$, characteristic $(k_m)_{m=1}^2$ suffice [Lyons, 2013].

38th Conference on Neural Information Processing Systems (NeurIPS 2024).

HSIC has been deployed successfully in numerous contexts, including independence testing in batch [Gretton et al., 2008, Wehbe and Ramdas, 2015, Bilodeau and Nangue, 2017, Górecki et al., 2018, Pfister et al., 2018, Albert et al., 2022, Shekhar et al., 2023] and streaming [Podkopaev et al., 2023] settings, feature selection [Camps-Valls et al., 2010, Song et al., 2012, Yamada et al., 2014, Wang et al., 2022] with applications in biomarker detection [Climente-González et al., 2019] and wind power prediction [Bouche et al., 2023], clustering [Song et al., 2007, Climente-González et al., 2019], and causal discovery [Mooij et al., 2016, Pfister et al., 2018, Chakraborty and Zhang, 2019, Schölkopf et al., 2021, Kalinke and Szabó, 2023]. In addition, HSIC has recently found successful applications in sensitivity analysis [Veiga, 2015, Freitas Gustavo et al., 2023, Fellmann et al., 2024, Herrando-Pérez and Saltré, 2024], in the context of uncertainty quantification [Stenger et al., 2020], for the analysis of data augmentation methods for brain tumor detection [Anaya-Isaza and Mera-Jiménez, 2022], and that of multimodal neural networks trained on neuroimaging data [Fedorov et al., 2024].

Many estimators for HSIC exist. The classical ones rely on U-statistics or V-statistics [Gretton et al., 2005, Quadrianto et al., 2009, Pfister et al., 2018] and are known to converge at a rate of $\mathcal{O}_P\left(n^{-1/2}\right)$. In fact, the V-statistic-based estimators are obtained by replacing the population kernel mean embedding with its empirical counterpart; estimating the mean embedding can be carried out at a speed $\mathcal{O}_P\left(n^{-1/2}\right)$ [Smola et al., 2007, Theorem 2], which implies that HSIC can be estimated at the same rate. Existing approximations such as Nyström HSIC [Kalinke and Szabó, 2023], also achieve this rate under the assumption of an appropriate rate of decay of the effective dimension. While all of these upper bounds match asymptotically, it is not known whether HSIC can be estimated at a faster rate, that is, whether the upper bound of $\mathcal{O}_P\left(n^{-1/2}\right)$ is optimal in the minimax sense, or if designing estimators achieving better rates is possible. Lower bounds for the related MMD are known [Tolstikhin et al., 2016], but the existing analysis considers radial kernels and relies on independent Gaussian distributions. Radial kernels are a special case of the more general class of translation-invariant kernels that we consider.[1] The reliance on independent Gaussian distributions renders the analysis of Tolstikhin et al. [2016] inapplicable for HSIC estimation. We tackle both of these severe restrictions in the present article.

We make the following **contributions**.

- We establish the minimax lower bound $\mathcal{O}\left(n^{-1/2}\right)$ of HSIC estimation with $M \geq 2$ components on $\mathbb{R}^d$ with continuous bounded translation-invariant characteristic kernels. As this lower bound matches the known upper bounds of the existing "classical" U-statistic and V-statistic-based estimators, and that of the Nyström HSIC estimator, our result settles their minimax optimality.

- Specifically, our result also implies the minimax lower bound of $\mathcal{O}\left(n^{-1/2}\right)$ for the estimation of the cross-covariance operator, which can be further specialized to get back the minimax result [Zhou et al., 2019, Theorem 5] on the estimation of the covariance operator.

The paper is structured as follows. Notations are introduced in Section 2. Section 3 is dedicated to our main result on the minimax rate of HSIC estimation on $\mathbb{R}^d$, with proof presented in Section 4. An auxiliary result on the Kullback-Leibler divergence is shown in Appendix A.

## 2 Notations

In this section, we introduce a few notations $\mathbb{N}_{>0}$, $[M]$, $\mathbf{I}_n$, $\mathbf{0}_n$, $\mathbf{1}_n$, $\mathbf{A}^\mathsf{T}$, $\langle \mathbf{v}, \mathbf{w} \rangle$, $\|\mathbf{v}\|_{\mathbb{R}^d}$, $\mathrm{bdiag}\left(\mathbf{M}_1, \ldots, \mathbf{M}_N\right)$, $|\mathbf{A}|$, $\mathcal{M}_1^+\left(\mathbb{R}^d\right)$, $\psi_{\mathbb{P}}$, $\mathrm{KL}(\mathbb{P}||\mathbb{Q})$, $L^2\left(\mathbb{R}^d, \Lambda\right)$, $\|f\|_{L^2(\mathbb{R}^d, \Lambda)}$, $\mathrm{supp}(\Lambda)$, $\mathcal{H}_k$, $\phi_k$, $k$, $\mu_k$, $\mathrm{MMD}_k$, $\otimes_{m=1}^M \mathcal{H}_{k_m}$, $\otimes_{m=1}^M k_m$, $\mathbb{P}_m$, $\otimes_{m=1}^M \mathbb{P}_m$, $\mathbb{P}^n$, $\mathcal{O}_P\left(r_n\right)$, $\mathcal{O}(a_n)$, $a_n \asymp b_n$, $\mathrm{HSIC}_k$, and $C_X$. Throughout the paper we consider random variables, probability measures, and kernels on $\mathbb{R}^d$.

For $M \in \mathbb{N}_{>0} := \{1, 2, \ldots\}$, let $[M] := \{1, \ldots, M\}$. Denote by $\mathbf{I}_n$ the $n \times n$-sized identity matrix and by $\mathbf{0}_n = (0, \ldots, 0)^\mathsf{T} \in \mathbb{R}^n$ (resp. $\mathbf{1}_n = (1, \ldots, 1)^\mathsf{T} \in \mathbb{R}^n$) a column vector of zeros (resp. ones). The transpose of a matrix $\mathbf{A} \in \mathbb{R}^{d_1 \times d_2}$ is written as $\mathbf{A}^\mathsf{T} \in \mathbb{R}^{d_2 \times d_1}$. For $\mathbf{v}, \mathbf{w} \in \mathbb{R}^d$, $\langle \mathbf{v}, \mathbf{w} \rangle = \mathbf{v}^\mathsf{T} \mathbf{w}$ stands for their Euclidean inner product; $\|\mathbf{v}\|_{\mathbb{R}^d} = \sqrt{\langle \mathbf{v}, \mathbf{v} \rangle}$ is the associated Euclidean norm.

---

[1]The family of radial kernels encompasses, for example, Gaussians, mixtures of Gaussians, inverse multi-quadratics, and Matérn kernels; the Laplace kernel is translation-invariant but not radial (with respect to the traditionally-chosen Euclidean norm $\|\cdot\|_{\mathbb{R}^d}$).

bdiag $(\mathbf{M}_1, \ldots, \mathbf{M}_N)$ forms a block-diagonal matrix from its arguments $(\mathbf{M}_n)_{n=1}^N$ $(\mathbf{M}_n \in \mathbb{R}^{d_n \times d_n}$, $n \in [N])$ and $|\mathbf{A}|$ denotes the determinant of a matrix $\mathbf{A} \in \mathbb{R}^{d \times d}$.

The set of Borel probability measures on $\mathbb{R}^d$ is denoted by $\mathcal{M}_1^+(\mathbb{R}^d)$. For a random variable $X \sim \mathbb{P} \in \mathcal{M}_1^+(\mathbb{R}^d)$, we denote its characteristic function by $\psi_{\mathbb{P}}(\boldsymbol{\omega}) = \mathbb{E}_{X \sim \mathbb{P}}[e^{i\langle \boldsymbol{\omega}, X \rangle}]$ with $\boldsymbol{\omega} \in \mathbb{R}^d$ and $i = \sqrt{-1}$. Let $\mathbb{P}, \mathbb{Q} \in \mathcal{M}_1^+(\mathbb{R}^d)$, assume that $\mathbb{P}$ is absolutely continuous w.r.t. $\mathbb{Q}$, and let $\frac{d\mathbb{P}}{d\mathbb{Q}}$ denote the corresponding Radon-Nikodym derivative (of $\mathbb{P}$ w.r.t. $\mathbb{Q}$). Then, the Kullback-Leibler divergence of $\mathbb{P}$ and $\mathbb{Q}$ is defined as $\text{KL}(\mathbb{P}||\mathbb{Q}) := \int_{\mathbb{R}^d} \log\left(\frac{d\mathbb{P}}{d\mathbb{Q}}(\mathbf{x})\right) d\mathbb{P}(\mathbf{x})$.

Given a measure space $(\mathbb{R}^d, \mathcal{B}(\mathbb{R}^d), \Lambda)$, we denote by $L^2(\mathbb{R}^d, \Lambda) := L^2(\mathbb{R}^d, \mathcal{B}(\mathbb{R}^d), \Lambda)$ the Hilbert space of (equivalence classes of) measurable functions $f: (\mathbb{R}^d, \mathcal{B}(\mathbb{R}^d)) \to (\mathbb{R}, \mathcal{B}(\mathbb{R}))$ for which $\|f\|_{L^2(\mathbb{R}^d, \Lambda)}^2 := \int_{\mathbb{R}^d} |f(\mathbf{x})|^2 d\Lambda(\mathbf{x}) < \infty$. The support of a probability measure $\Lambda \in \mathcal{M}_1^+(\mathbb{R}^d)$ denoted by $\text{supp}(\Lambda)$ is the subset of $\mathbb{R}^d$ for which every open neighborhood of $\mathbf{x} \in \mathbb{R}^d$ has positive measure [Cohn, 2013, p. 207].

A function $k: \mathbb{R}^d \times \mathbb{R}^d \to \mathbb{R}$ is called a kernel if there exists a Hilbert space $\mathcal{H}$ and a feature map $\phi: \mathbb{R}^d \to \mathcal{H}$ such that $k(\mathbf{x}, \mathbf{x}') = \langle \phi(\mathbf{x}), \phi(\mathbf{x}') \rangle_{\mathcal{H}}$ for all $\mathbf{x}, \mathbf{x}' \in \mathbb{R}^d$. A Hilbert space of functions $h: \mathbb{R}^d \to \mathbb{R}$ is an RKHS $\mathcal{H}_k$ associated to a kernel $k: \mathbb{R}^d \times \mathbb{R}^d \to \mathbb{R}$ if $k(\cdot, \mathbf{x}) \in \mathcal{H}_k$ and $\langle h, k(\cdot, \mathbf{x}) \rangle_{\mathcal{H}_k} = h(\mathbf{x})$ for all $\mathbf{x} \in \mathbb{R}^d$ and $h \in \mathcal{H}_k$.[2] In this work, we assume all kernels to be measurable and bounded.[3] The function $\phi_k(\mathbf{x}) := k(\cdot, \mathbf{x})$ is the canonical feature map, and $k(\mathbf{x}, \mathbf{x}') = \langle k(\cdot, \mathbf{x}), k(\cdot, \mathbf{x}') \rangle_{\mathcal{H}_k} = \langle \phi_k(\mathbf{x}), \phi_k(\mathbf{x}') \rangle_{\mathcal{H}_k}$ for all $\mathbf{x}, \mathbf{x}' \in \mathbb{R}^d$. A function $\kappa: \mathbb{R}^d \to \mathbb{R}$ is called positive definite if $\sum_{i,j \in [n]} c_i c_j \kappa(\mathbf{x}_i - \mathbf{x}_j) \geq 0$ for all $n \in \mathbb{N}_{>0}$, $\mathbf{c} = (c_i)_{i=1}^n \in \mathbb{R}^n$, and $\{\mathbf{x}_i\}_{i=1}^n \subset \mathbb{R}^d$. A kernel $k: \mathbb{R}^d \times \mathbb{R}^d \to \mathbb{R}$ is said to be translation-invariant if there exists a positive definite function $\kappa: \mathbb{R}^d \to \mathbb{R}$ such that $k(\mathbf{x}, \mathbf{x}') = \kappa(\mathbf{x} - \mathbf{x}')$ for all $\mathbf{x}, \mathbf{x}' \in \mathbb{R}^d$. By Bochner's theorem [Wendland, 2005, Theorem 6.6] (recalled in Theorem B.1) for a continuous bounded translation-invariant kernel $k: \mathbb{R}^d \times \mathbb{R}^d \to \mathbb{R}$ there exists a finite non-negative Borel measure $\Lambda_k$ such that

$$k(\mathbf{x}, \mathbf{y}) = \int_{\mathbb{R}^d} e^{-i\langle \mathbf{x} - \mathbf{y}, \boldsymbol{\omega} \rangle} d\Lambda_k(\boldsymbol{\omega}) \tag{1}$$

for all $\mathbf{x}, \mathbf{y} \in \mathbb{R}^d$. The (kernel) mean embedding of a probability measure $\mathbb{P} \in \mathcal{M}_1^+(\mathbb{R}^d)$ is

$$\mu_k(\mathbb{P}) = \int_{\mathbb{R}^d} \phi_k(\mathbf{x}) d\mathbb{P}(\mathbf{x}) \in \mathcal{H}_k,$$

where the integral is meant in Bochner's sense [Diestel and Uhl, 1977, Chapter II.2]; the boundedness of $k$ ensures that it is well-defined. For $\mathbb{P}, \mathbb{Q} \in \mathcal{M}_1^+(\mathbb{R}^d)$ one can define the (semi-)metric called maximum mean discrepancy [Smola et al., 2007, Gretton et al., 2012] as

$$\text{MMD}_k(\mathbb{P}, \mathbb{Q}) = \|\mu_k(\mathbb{P}) - \mu_k(\mathbb{Q})\|_{\mathcal{H}_k}.$$

If the mean embedding $\mu_k$ is injective, MMD is a metric and the kernel $k$ is called characteristic [Fukumizu et al., 2008, Sriperumbudur et al., 2010, Szabó and Sriperumbudur, 2018].

Let $\mathbb{R}^d = \times_{m=1}^M \mathbb{R}^{d_m}$ $(d = \sum_{m=1}^M d_m)$ and assume that each domain $\mathbb{R}^{d_m}$ is equipped with a kernel $k_m: \mathbb{R}^{d_m} \times \mathbb{R}^{d_m} \to \mathbb{R}$ with associated RKHS $\mathcal{H}_{k_m}$ $(m \in [M])$. The tensor product Hilbert space of $(\mathcal{H}_{k_m})_{m=1}^M$ is denoted by $\otimes_{m=1}^M \mathcal{H}_{k_m}$; it is an RKHS [Berlinet and Thomas-Agnan, 2004, Theorem 13] with the tensor product kernel $k = \otimes_{m=1}^M k_m: \mathbb{R}^d \times \mathbb{R}^d \to \mathbb{R}$ defined by

$$k\left((\mathbf{x}_m)_{m=1}^M, (\mathbf{x}_m')_{m=1}^M\right) = \prod_{m \in [M]} k_m(\mathbf{x}_m, \mathbf{x}_m') \quad \text{for all} \quad \mathbf{x}_m, \mathbf{x}_m' \in \mathbb{R}^{d_m}, m \in [M].$$

The kernel $k$ has the canonical feature map $\phi_k\left((\mathbf{x}_m)_{m=1}^M\right) = \otimes_{m=1}^M \phi_{k_m}(\mathbf{x}_m) \in \otimes_{m=1}^M \mathcal{H}_{k_m} =: \mathcal{H}_k$ $(\mathbf{x}_m \in \mathbb{R}^{d_m}, m \in [M])$. Let $X = (X_m)_{m=1}^M$ be a random variable taking values in $\mathbb{R}^d$

---

[2]For fixed $\mathbf{x} \in \mathbb{R}^d$, the function $k(\cdot, \mathbf{x}): \mathbb{R}^d \to \mathbb{R}$ means $\mathbf{x}' \mapsto k(\mathbf{x}', \mathbf{x})$.

[3]Boundedness of the kernel, that is, $\sup_{\mathbf{x}, \mathbf{x}' \in \mathbb{R}^d} k(\mathbf{x}, \mathbf{x}') < \infty$, implies boundedness of the feature map, that is, $\sup_{\mathbf{x} \in \mathbb{R}^d} \|\phi_k(\mathbf{x})\|_{\mathcal{H}_k} < \infty$ (and vice versa); it is also equivalent to $\sup_{\mathbf{x} \in \mathbb{R}^d} k(\mathbf{x}, \mathbf{x}) < \infty$.

with joint distribution $\mathbb{P} \in \mathcal{M}_1^+ \left(\mathbb{R}^d\right)$ and marginal distributions $\mathbb{P}_m \in \mathcal{M}_1^+ \left(\mathbb{R}^{d_m}\right)$ $(m \in [M];$ $d = \sum_{m=1}^M d_m)$. We write $\otimes_{m=1}^M \mathbb{P}_m \in \mathcal{M}_1^+ \left(\mathbb{R}^d\right)$ for the product of measures $\mathbb{P}_m$ $(m \in [M])$. Specifically, $\mathbb{P}^n := \otimes_{i=1}^n \mathbb{P} \in \mathcal{M}_1^+ \left(\left(\mathbb{R}^d\right)^n\right)$ denotes the $n$-fold product of $\mathbb{P}$. For a sequence of real-valued random variables $(X_n)_{n=1}^\infty$ and a sequence $(r_n)_{n=1}^\infty$ $(r_n > 0$ for all $n)$, $X_n = \mathcal{O}_P\left(r_n\right)$ denotes that $\frac{X_n}{r_n}$ is bounded in probability. For positive sequences $(a_n)_{n=1}^\infty$ and $(b_n)_{n=1}^\infty$, $b_n = \mathcal{O}(a_n)$ if there exist constants $C > 0$ and $n_0 \in \mathbb{N}_{>0}$ such that $b_n \leq C a_n$ for all $n \geq n_0$; $a_n \asymp b_n$ if $a_n = \mathcal{O}\left(b_n\right)$ and $b_n = \mathcal{O}\left(a_n\right)$. One can define our quantity of interest, the Hilbert-Schmidt independence criterion (HSIC; [Gretton et al., 2005, Quadrianto et al., 2009, Pfister et al., 2018, Szabó and Sriperumbudur, 2018]), as

$$\mathrm{HSIC}_k(\mathbb{P}) = \mathrm{MMD}_k \left(\mathbb{P}, \otimes_{m=1}^M \mathbb{P}_m\right) = \|C_X\|_{\mathcal{H}_k},$$
$$C_X = \mu_k(\mathbb{P}) - \mu_k \left(\otimes_{m=1}^M \mathbb{P}_m\right) \in \mathcal{H}_k, \tag{2}$$

and $C_X$ denotes the centered cross-covariance operator.

## 3 Results

This section is dedicated to our results: The minimax lower bound for the estimation of $\mathrm{HSIC}_k(\mathbb{P})$, where $k$ is a product of continuous bounded translation-invariant characteristic kernels is given in Theorem 1(ii). For the specific case where $k$ is a product of Gaussian kernels (stated in Theorem 1(i)), the constant in the lower bound is made explicit. Theorem 1(ii) also helps to establish a lower bound on the estimation of the cross-covariance operator (Corollary 1).

Before presenting our results, we recall the framework of minimax estimation [Tsybakov, 2009] adapted to our setting. Let $\hat{F}_n$ denote any estimator of $\mathrm{HSIC}_k(\mathbb{P})$ based on $n$ i.i.d. samples from $\mathbb{P}$. A sequence $(\xi_n)_{n=1}^\infty$ $(\xi_n > 0$ for all $n)$ is said to be a lower bound of HSIC estimation w.r.t. a class $\mathcal{P}$ of Borel probability measures on $\mathbb{R}^d$ if there exists a constant $c > 0$ such that

$$\inf_{\hat{F}_n} \sup_{\mathbb{P} \in \mathcal{P}} \mathbb{P}^n \left\{ \xi_n^{-1} \left| \mathrm{HSIC}_k(\mathbb{P}) - \hat{F}_n \right| \geq c \right\} > 0. \tag{3}$$

If a specific estimator of HSIC $\tilde{F}_n$ has an upper bound that matches $(\xi_n)_{n=1}^\infty$ up to constants, that is,

$$\left| \mathrm{HSIC}_k(\mathbb{P}) - \tilde{F}_n \right| = \mathcal{O}_P\left(\xi_n\right), \tag{4}$$

then $\tilde{F}_n$ is called minimax optimal.

We use Le Cam's method [Le Cam, 1973, Tsybakov, 2009] (recalled in Theorem B.5) to obtain bounds as in (3); estimators of HSIC achieving the bounds in (4) with $\xi_n = n^{-1/2}$ are quoted in the introduction. The key to the application of the method is to show that there exist $\alpha > 0$ and $n_0 \in \mathbb{N}_{>0}$ such that for all $n \geq n_0$ one can find an adversarial pair of distributions $(\mathbb{P}_{\theta_0}, \mathbb{P}_{\theta_1}) = (\mathbb{P}_{\theta_0}(n), \mathbb{P}_{\theta_1}(n)) \in \mathcal{P} \times \mathcal{P}$ and $s_n > 0$ for which

1. $\mathrm{KL}\left(\mathbb{P}_{\theta_1}^n || \mathbb{P}_{\theta_0}^n\right) \leq \alpha$, in other words, the corresponding $n$-fold product measures must be similar in the sense of Kullback-Leibler divergence, but

2. $|\mathrm{HSIC}_k(\mathbb{P}_{\theta_1}) - \mathrm{HSIC}_k(\mathbb{P}_{\theta_0})| \geq 2s_n$, that is, their corresponding values of HSIC must be dissimilar.

In this case, $\inf_{\hat{F}_n} \sup_{\mathbb{P} \in \mathcal{P}} \mathbb{P}^n \left\{ \left| \mathrm{HSIC}_k(\mathbb{P}) - \hat{F}_n \right| \geq s_n \right\} \geq \max\left(\frac{e^{-\alpha}}{4}, \frac{1-\sqrt{\alpha/2}}{2}\right)$ for all $n \geq n_0$; hence to establish the minimax optimality of existing estimators w.r.t. their known upper bounds, it is sufficient to find adversarial pairs $\{(\mathbb{P}_{\theta_0}(n), \mathbb{P}_{\theta_1}(n))\}_{n \geq n_0}$ that satisfy 1. for some positive constant $\alpha$ and also fulfill 2. with $s_n \asymp n^{-1/2}$.

The proof of the first part of our statement relies on the following Lemma 1 which yields the analytical value of $\mathrm{HSIC}_k\left(\mathcal{N}(\boldsymbol{\mu}, \boldsymbol{\Sigma})\right)$, where $k = \otimes_{m=1}^M k_m$ is the product of Gaussian kernels $k_m$ $(m \in [M])$ and $\mathcal{N}(\boldsymbol{\mu}, \boldsymbol{\Sigma})$ denotes the multivariate normal distribution with mean $\boldsymbol{\mu} \in \mathbb{R}^d$ and covariance matrix $\boldsymbol{\Sigma} \in \mathbb{R}^{d \times d}$.

**Lemma 1** (Analytical value of HSIC for the Gaussian setting). *Let us consider the Gaussian kernel $k(\mathbf{x}, \mathbf{y}) = e^{-\frac{\gamma}{2}\|\mathbf{x}-\mathbf{y}\|^2_{\mathbb{R}^d}}$ ($\gamma > 0$, $\mathbf{x}, \mathbf{y} \in \mathbb{R}^d$) and Gaussian random variable $X = (X_m)_{m=1}^M \sim \mathcal{N}(\mathbf{m}, \mathbf{\Sigma}) =: \mathbb{P}$, where $X_m \in \mathbb{R}^{d_m}$ ($m \in [M]$), $\mathbf{m} = (\mathbf{m}_m)_{m=1}^M \in \mathbb{R}^d$, $\mathbf{\Sigma} = [\mathbf{\Sigma}_{i,j}]_{i,j\in[M]} \in \mathbb{R}^{d\times d}$, $\mathbf{\Sigma}_{i,j} \in \mathbb{R}^{d_i \times d_j}$, and $d = \sum_{m\in[M]} d_m$. In this case, with $\mathbf{\Sigma}_1 = \mathbf{\Sigma}$ and $\mathbf{\Sigma}_2 = \mathrm{bdiag}(\mathbf{\Sigma}_{1,1}, \ldots, \mathbf{\Sigma}_{M,M})$, we have*

$$\mathrm{HSIC}^2_k(\mathbb{P}) = \frac{1}{|2\gamma\mathbf{\Sigma}_1 + \mathbf{I}_d|^{\frac{1}{2}}} + \frac{1}{|2\gamma\mathbf{\Sigma}_2 + \mathbf{I}_d|^{\frac{1}{2}}} - \frac{2}{|\gamma\mathbf{\Sigma}_1 + \gamma\mathbf{\Sigma}_2 + \mathbf{I}_d|^{\frac{1}{2}}}.$$

In this work, we focus on continuous bounded translation-invariant kernels, which are fully characterized by Bochner's theorem [Wendland, 2005, Theorem 6.6]; the theorem states that a function on $\mathbb{R}^d$ is positive definite if and only if it is the Fourier transform of a finite nonnegative measure.[4] We use this description to obtain our main result, which is as follows.

**Theorem 1** (Lower bound for HSIC estimation on $\mathbb{R}^d$). *Let $\mathcal{P}$ be a class of Borel probability measures over $\mathbb{R}^d$ containing the $d$-dimensional Gaussian distributions. Let $d = \sum_{m\in[M]} d_m$ and $\hat{F}_n$ denote any estimator of $\mathrm{HSIC}_k(\mathbb{P})$ with $n \geq 2 =: n_0$ i.i.d. samples from $\mathbb{P} \in \mathcal{P}$. Assume further that $k = \otimes_{m=1}^M k_m$ where either, for $m \in [M]$,*

*(i) the kernels $k_m : \mathbb{R}^{d_m} \times \mathbb{R}^{d_m} \to \mathbb{R}$ are Gaussian with common bandwidth parameter $\gamma > 0$ defined by $(\mathbf{x}_m, \mathbf{x}'_m) \mapsto e^{-\frac{\gamma}{2}\|\mathbf{x}_m - \mathbf{x}'_m\|^2_{\mathbb{R}^{d_m}}}$ ($\mathbf{x}_m, \mathbf{x}'_m \in \mathbb{R}^{d_m}$), or*

*(ii) the kernels $k_m : \mathbb{R}^{d_m} \times \mathbb{R}^{d_m} \to \mathbb{R}$ are continuous bounded translation-invariant characteristic kernels.*

*Then, for any $n \geq n_0$, it holds that*

$$\inf_{\hat{F}_n} \sup_{\mathbb{P}\in\mathcal{P}} \mathbb{P}^n \left\{ \left| \mathrm{HSIC}_k(\mathbb{P}) - \hat{F}_n \right| \geq \frac{c}{\sqrt{n}} \right\} \geq \frac{1 - \sqrt{\frac{5}{8}}}{2},$$

*with (i) the constant $c = \frac{\gamma}{2(2\gamma+1)^{\frac{d}{4}+1}} > 0$ (depending on $\gamma$ and $d$ only) in the first case, or (ii) some constant $c > 0$ in the second case.*

We note that while Theorem 1(ii) applies to the more general class of translation-invariant kernels, we include Theorem 1(i) as it makes the constant $c$ explicit.

The following corollary allows to recover the recent lower bound on the estimation of the covariance operator by Zhou et al. [2019, Theorem 5] as a special case that we detail in Remark 1(e).

**Corollary 1** (Lower bound on cross-covariance operator estimation). *In the setting of Theorem 1(ii), let $\hat{F}_n$ denote any estimator of the centered cross-covariance operator $C_X \in \mathcal{H}_k$ defined in (2) with $n \geq 2 =: n_0$ i.i.d. samples from $\mathbb{P} \in \mathcal{P}$. Then, for any $n \geq n_0$, it holds that*

$$\inf_{\hat{F}_n} \sup_{\mathbb{P}\in\mathcal{P}} \mathbb{P}^n \left\{ \left\| C_X - \hat{F}_n \right\|_{\mathcal{H}_k} \geq \frac{c}{\sqrt{n}} \right\} \geq \frac{1 - \sqrt{\frac{5}{8}}}{2},$$

*for some constant $c > 0$.*

**Remark 1.**

*(a) **Validness of HSIC.** Though generally the characteristic property of $(k_m)_{m=1}^M$-s is not enough [Szabó and Sriperumbudur, 2018, Example 2] for $M > 2$ to ensure the $\mathcal{I}$-characteristic property of $k = \otimes_{m=1}^M k_m$ (in other words, that $\mathrm{HSIC}_k(\mathbb{P}) = 0$ iff. $\mathbb{P} = \otimes_{m=1}^M \mathbb{P}_m$), on $\mathbb{R}^d$ under the imposed continuous bounded translation-invariant assumption (i) $k$ being characteristic, (ii) $k$ being $\mathcal{I}$-characteristic, and (iii) $(k_m)_{m=1}^M$-s being characteristic are equivalent (Theorem B.4).*

*(b) **Minimax optimality of existing HSIC estimators.** The lower bounds in Theorem 1 asymptotically match the known upper bounds of the U-statistic and V-statistic-based estimators of $\xi_n = n^{-1/2}$.*

---

[4]We note that for many translation-invariant kernels, the corresponding spectral measures are known [Sriperumbudur et al., 2010, Table 2].

*The Nyström-based HSIC estimator achieves the same rate under an appropriate decay of the eigenspectrum of the respective covariance operator. Hence, Theorem 1 implies the optimality of these estimators on $\mathbb{R}^d$ with continuous bounded translation-invariant characteristic kernels in the minimax sense.*

*(c)* ***Difference compared to Tolstikhin et al. [2016] (minimax MMD estimation).*** *We note that a lower bound for the related $MMD_k$ exists. However, the adversarial distribution pair $(\mathbb{P}_{\theta_1}, \mathbb{P}_{\theta_0})$ constructed by Tolstikhin et al. [2016, Theorem 1] to obtain the lower bound on MMD estimation has a product structure which implies that $|\mathrm{HSIC}_k(\mathbb{P}_{\theta_1}) - \mathrm{HSIC}_k(\mathbb{P}_{\theta_0})| = 0$ and hence it is not applicable in our case of HSIC; Tolstikhin et al. [2016, Theorem 2] with radial kernels has the same restriction.*

*(d)* ***Difference compared to Tolstikhin et al. [2017] (minimax mean embedding estimation).*** *The estimation of the mean embedding $\mu_k(\mathbb{P})$ is known to have a minimax rate of $\mathcal{O}\left(n^{-1/2}\right)$. But, this rate does not imply an optimal lower bound for the estimation of MMD as is evident from the two works [Tolstikhin et al., 2016, 2017]. The same conclusion holds for HSIC estimation.*

*(e)* ***Difference compared to Zhou et al. [2019] (minimax covariance operator estimation).*** *For the related problem of estimating the centered covariance operator*

$$C_{XX} = \int_{\mathbb{R}^d} \left(\phi_k(x) - \mu_k(\mathbb{P})\right) \otimes \left(\phi_k(x) - \mu_k(\mathbb{P})\right) \mathrm{d}\mathbb{P}(x) \in \mathcal{H}_k \otimes \mathcal{H}_k,$$

*Zhou et al. [2019, Theorem 5] give the lower bound*

$$\inf_{\hat{F}_n} \sup_{\mathbb{P} \in \mathcal{P}} \mathbb{P}^n \left\{ \left\| C_{XX} - \hat{F}_n \right\|_{\mathcal{H}_k \otimes \mathcal{H}_k} \geq \frac{c}{\sqrt{n}} \right\} \geq 1/8$$

*in the same setting as in Theorem 1(ii), where $\hat{F}_n$ is any estimator of the centered covariance $C_{XX}$, and $c$ is a positive constant. By noting that the centered covariance is the centered cross-covariance of a random variable with itself, Corollary 1 recovers their result.*

The next section contains our proofs.

## 4  Proofs

This section is dedicated to our proofs. We present the proof of Lemma 1 in Section 4.1, that of Theorem 1 in Section 4.2, and that of Corollary 1 in Section 4.3.

### 4.1  Proof of Lemma 1

As

$$\begin{aligned}
\mathrm{HSIC}_k^2(\mathbb{P}) = \mathrm{MMD}_k^2(\mathbb{P}, \mathbb{Q}) &= \|\mu_k(\mathbb{P}) - \mu_k(\mathbb{Q})\|_{\mathcal{H}_k}^2 \\
&= \langle \mu_k(\mathbb{P}), \mu_k(\mathbb{P}) \rangle_{\mathcal{H}_k} + \langle \mu_k(\mathbb{Q}), \mu_k(\mathbb{Q}) \rangle_{\mathcal{H}_k} - 2\langle \mu_k(\mathbb{P}), \mu_k(\mathbb{Q}) \rangle_{\mathcal{H}_k}
\end{aligned}$$

with $\mathbb{Q} = \otimes_{m=1}^M \mathbb{P}_m = \mathcal{N}(\mathbf{m}, \mathrm{bdiag}(\boldsymbol{\Sigma}_{1,1}, \ldots, \boldsymbol{\Sigma}_{M,M}))$, $\mathbb{P}_m = \mathcal{N}(\mathbf{m}_m, \boldsymbol{\Sigma}_{m,m})$, it is sufficient to be able to compute $\langle \mu_k(\mathbb{P}), \mu_k(\mathbb{Q}) \rangle_{\mathcal{H}_k}$-type quantities with $\mathbb{P} = \mathcal{N}(\mathbf{m}_1, \boldsymbol{\Sigma}_1)$ and $\mathbb{Q} = \mathcal{N}(\mathbf{m}_2, \boldsymbol{\Sigma}_2)$. One can show [Muandet et al., 2011, Table 1] that $\langle \mu_k(\mathbb{P}), \mu_k(\mathbb{Q}) \rangle_{\mathcal{H}_k} = \frac{e^{-\frac{1}{2}(\mathbf{m}_1 - \mathbf{m}_2)^\top \left(\boldsymbol{\Sigma}_1 + \boldsymbol{\Sigma}_2 + \gamma^{-1}\mathbf{I}_d\right)^{-1}(\mathbf{m}_1 - \mathbf{m}_2)}}{|\gamma\boldsymbol{\Sigma}_1 + \gamma\boldsymbol{\Sigma}_2 + \mathbf{I}_d|^{\frac{1}{2}}}$. Using this fact and that $\mathbf{m} = \mathbf{m}_1 = \mathbf{m}_2$, the result follows.

### 4.2  Proof of Theorem 1

The setup and the upper bound on $\mathrm{KL}(\mathbb{P}_{\theta_1}^n || \mathbb{P}_{\theta_0}^n)$ agree for (i) and (ii) but the methods that we use to lower bound $|\mathrm{HSIC}_k(\mathbb{P}_{\theta_1}) - \mathrm{HSIC}_k(\mathbb{P}_{\theta_0})|$ differ. We structure the proof accordingly and present the overlapping part before we branch out into (i) and (ii). Both parts of the statement rely on Le Cam's method, which we state as Theorem B.5 for self-completeness.

To construct the adversarial pair, we consider a class $\mathcal{G}$ of Gaussian distributions over $\mathbb{R}^d$ such that every element $\mathcal{N}(\boldsymbol{\mu}, \boldsymbol{\Sigma}) \in \mathcal{G}$, with

$$\boldsymbol{\Sigma} = \boldsymbol{\Sigma}(i,j,\rho) = \begin{bmatrix} 1 & \cdots & 0 & 0 & \cdots & 0 \\ \vdots & \ddots & \vdots & \vdots & & \vdots \\ 0 & \cdots & 1 & \rho & \cdots & 0 \\ 0 & \cdots & \rho & 1 & \cdots & 0 \\ \vdots & & \vdots & \vdots & \ddots & \vdots \\ 0 & \cdots & 0 & 0 & \cdots & 1 \end{bmatrix} \in \mathbb{R}^{d \times d}, \tag{5}$$

and (fixed) $i = d_1$, $j = d_1 + 1$, $\rho \in (-1,1)$. In other words, $\boldsymbol{\Sigma}$ is essentially the $d$-dimensional matrix $\boldsymbol{I}_d$ except for the $(i,j)$ and $(j,i)$ entry; both entries are identical to $\rho$, and they specify the correlation of the respective coordinates. This family of distributions is indexed by a tuple $(\boldsymbol{\mu}, \rho) \in \mathbb{R}^d \times (-1,1) =: \mathcal{A}$ and, for $a \in \mathcal{A}$, we write $\mathbb{P}_a$ for the associated distribution. To bring ourselves into the setting of Theorem B.5, we fix $n \in \mathbb{N}_{>0}$, choose $\mathcal{X} = (\mathbb{R}^d)^n$, set $\Theta = \{\theta_a := \mathrm{HSIC}_k(\mathbb{P}_a) : a \in \mathcal{A}\}$, $\mathcal{P}_\Theta = \{\mathbb{P}_a^n : a \in \mathcal{A}\} = \{\mathbb{P}_a^n : \theta_a \in \Theta\}$, and use the metric $(x,y) \mapsto |x-y|$ for $x, y \in \mathbb{R}$. Hence, the data $D \sim \mathbb{P}_\theta \in \mathcal{P}_\Theta$. For brevity, let $F : \mathcal{A} \to \mathbb{R}$ stand for $a \mapsto \mathrm{HSIC}_k(\mathbb{P}_a)$, and let $\hat{F}_n$ stand for the corresponding estimator based on $n$ samples.

As $\mathcal{G} \subseteq \mathcal{P}$, it holds for every positive $s$ that

$$\sup_{\mathbb{P} \in \mathcal{P}} \mathbb{P}^n \left\{ \left| \mathrm{HSIC}_k(\mathbb{P}) - \hat{F}_n \right| \geq s \right\} \geq \sup_{\mathbb{P} \in \mathcal{G}} \mathbb{P}^n \left\{ \left| \mathrm{HSIC}_k(\mathbb{P}) - \hat{F}_n \right| \geq s \right\}.$$

Let $\mathbb{P}_{\theta_0} = \mathcal{N}(\boldsymbol{\mu}_0, \boldsymbol{\Sigma}_0)$ and $\mathbb{P}_{\theta_1} = \mathcal{N}(\boldsymbol{\mu}_1, \boldsymbol{\Sigma}_1)$ with

$$\boldsymbol{\mu}_0 = \mathbf{0}_d \in \mathbb{R}^d, \qquad\qquad \boldsymbol{\Sigma}_0 = \boldsymbol{\Sigma}(d_1, d_1 + 1, 0) = \mathbf{I}_d \in \mathbb{R}^{d \times d},$$
$$\boldsymbol{\mu}_1 = \frac{1}{\sqrt{dn}} \mathbf{1}_d \in \mathbb{R}^d, \qquad\qquad \boldsymbol{\Sigma}_1 = \boldsymbol{\Sigma}(d_1, d_1 + 1, \rho_n) \in \mathbb{R}^{d \times d},$$

where $\rho_n \in (-1,1)$ will be chosen appropriately later.[5] We now proceed to upper bound $\mathrm{KL}\left(\mathbb{P}_{\theta_1}^n \| \mathbb{P}_{\theta_0}^n\right)$ and lower bound $|F(\theta_1) - F(\theta_0)|$.

**Upper bound for KL divergence** Lemma A.1 implies that with $\rho_n^2 = \frac{1}{n}$, one has the bound $\mathrm{KL}\left(\mathbb{P}_{\theta_1}^n \| \mathbb{P}_{\theta_0}^n\right) \leq \alpha := \frac{5}{4}$ for $n \geq 2 =: n_0$.

**Lower bound (i): Gaussian kernels.** Recall that the considered kernel is $k(\mathbf{x}, \mathbf{y}) = e^{-\frac{\gamma}{2} \|\mathbf{x} - \mathbf{y}\|_{\mathbb{R}^d}^2}$ ($\gamma > 0$). The idea of the proof is as follows.

1. We express $|F(\theta_1) - F(\theta_0)|$ in closed form as a function of $\gamma$, $\rho_n$, and $d$.

2. Using the analytical form obtained in the 1st step, we construct the lower bound.

This is what we detail next.

- **Analytical form of $|F(\theta_1) - F(\theta_0)|$:** Using the fact that $\mathrm{HSIC}_k(\mathbb{P}_{\theta_0}) = 0$, we have that

$$\left| F(\theta_1) - \underbrace{F(\theta_0)}_{=0} \right|^2 = F^2(\theta_1) = \mathrm{HSIC}_k^2(\mathbb{P}_{\theta_1}) = \mathrm{MMD}_k^2\left(\mathcal{N}(\boldsymbol{\mu}_1, \boldsymbol{\Sigma}_1), \mathcal{N}(\boldsymbol{\mu}_1, \mathbf{I}_d)\right)$$
$$= \|\mu_k(\mathcal{N}(\boldsymbol{\mu}_1, \boldsymbol{\Sigma}_1)) - \mu_k(\mathcal{N}(\boldsymbol{\mu}_1, \mathbf{I}_d))\|_{\mathcal{H}_k}^2$$
$$= \underbrace{\langle \mu_k(\mathcal{N}(\boldsymbol{\mu}_1, \boldsymbol{\Sigma}_1)), \mu_k(\mathcal{N}(\boldsymbol{\mu}_1, \boldsymbol{\Sigma}_1)) \rangle_{\mathcal{H}_k}}_{(i)} + \underbrace{\langle \mu_k(\mathcal{N}(\boldsymbol{\mu}_1, \mathbf{I}_d)), \mu_k(\mathcal{N}(\boldsymbol{\mu}_1, \mathbf{I}_d)) \rangle_{\mathcal{H}_k}}_{(ii)}$$
$$- 2 \underbrace{\langle \mu_k(\mathcal{N}(\boldsymbol{\mu}_1, \boldsymbol{\Sigma}_1)), \mu_k(\mathcal{N}(\boldsymbol{\mu}_1, \mathbf{I}_d)) \rangle_{\mathcal{H}_k}}_{(iii)},$$

---

[5]Notice the dependence of $\mathbb{P}_{\theta_1}$ on $n$.

which we compute term-by-term with Lemma 1, and obtain

$$(i) = |2\gamma\boldsymbol{\Sigma}_1 + \mathbf{I}_d|^{-1/2} = \left[(2\gamma + 1)^{d-2}\left((2\gamma + 1)^2 - (2\gamma\rho_n)^2\right)\right]^{-1/2},$$

$$(ii) = |2\gamma\mathbf{I}_d + \mathbf{I}_d|^{-1/2} = \left[(2\gamma + 1)^d\right]^{-1/2},$$

$$(iii) = |\gamma\boldsymbol{\Sigma}_1 + \gamma\mathbf{I}_d + \mathbf{I}_d|^{-1/2} = \left[(2\gamma + 1)^{d-2}\left((2\gamma + 1)^2 - (\gamma\rho_n)^2\right)\right]^{-1/2}.$$

Combining (i), (ii), and (iii) yields that

$$\begin{aligned}
\text{HSIC}_k^2\left(\mathbb{P}_{\theta_1}\right) &= (i) + (ii) - 2(iii) \\
&= \left[(2\gamma + 1)^{d-2}\left((2\gamma + 1)^2 - (2\gamma\rho_n)^2\right)\right]^{-1/2} + \left[(2\gamma + 1)^d\right]^{-1/2} \\
&\quad - 2\left[(2\gamma + 1)^{d-2}\left((2\gamma + 1)^2 - (\gamma\rho_n)^2\right)\right]^{-1/2}.
\end{aligned}$$

- **Lower bound on** $|F(\theta_1) - F(\theta_0)|$: Next, we show that there exists $c > 0$ such that for any $n \in \mathbb{N}_{>0}$ it holds that $\text{HSIC}_k^2\left(\mathbb{P}_{\theta_1}\right) \geq \frac{c}{n}$.

  For $0 < x < \left(1 + \frac{1}{2\gamma}\right)^2$, let us consider the function

$$\begin{aligned}
f_c(x) &= \left[(2\gamma + 1)^{d-2}\left((2\gamma + 1)^2 - 4\gamma^2 x\right)\right]^{-1/2} + \left[(2\gamma + 1)^d\right]^{-1/2} \\
&\quad - 2\left[(2\gamma + 1)^{d-2}\left((2\gamma + 1)^2 - \gamma^2 x\right)\right]^{-1/2} - cx \\
&= \left[z^{d-2}\left(z^2 - 4\gamma^2 x\right)\right]^{-1/2} + \left(z^d\right)^{-1/2} - 2\left[z^{d-2}\left(z^2 - \gamma^2 x\right)\right]^{-1/2} - cx,
\end{aligned}$$

  with the shorthand $z := 2\gamma + 1$.[6] With this notation, $f_c(1/n) = \text{HSIC}_k^2\left(\mathbb{P}_{\theta_1}\right) - c/n$; our aim is to determine $c > 0$ such that $f_c(1/n) \geq 0$ for any positive integer $n$. To achieve this goal, notice that $f_c(0) = 0$, and

$$\begin{aligned}
f_c'(x) &= \frac{2\gamma^2 z^{d-2}}{\left[z^{d-2}\left(z^2 - 4x\gamma^2\right)\right]^{3/2}} - \frac{\gamma^2 z^{d-2}}{\left[z^{d-2}\left(z^2 - x\gamma^2\right)\right]^{3/2}} - c \\
&> \frac{2\gamma^2 z^{d-2}}{\left[z^{d-2}\left(z^2 - x\gamma^2\right)\right]^{3/2}} - \frac{\gamma^2 z^{d-2}}{\left[z^{d-2}\left(z^2 - x\gamma^2\right)\right]^{3/2}} - c = \frac{\gamma^2 z^{d-2}}{\left[z^{d-2}\left(z^2 - x\gamma^2\right)\right]^{3/2}} - c \\
&> \frac{\gamma^2 z^{d-2}}{\left(z^{d-2}z^2\right)^{3/2}} - c = \frac{\gamma^2}{z^2\sqrt{z^d}} - c = \frac{\gamma^2}{(2\gamma + 1)^2\sqrt{(2\gamma + 1)^d}} - c.
\end{aligned}$$

  Choosing now $c = \frac{\gamma^2}{(2\gamma+1)^2\sqrt{(2\gamma+1)^d}} > 0$, we have $f_c'(x) \geq 0$, so $f$ is a nondecreasing function.

  Note that $f_c(1/n) = \text{HSIC}_k^2\left(\mathbb{P}_{\theta_1}\right) - c/n \geq 0$, with $x = 1/n$ and $\left(1 + \frac{1}{2\gamma}\right)^{-2} < 1 \leq n < \infty$. By taking the positive square root, this means that

$$\text{HSIC}_k\left(\mathbb{P}_{\theta_1}\right) \geq \frac{\gamma}{(2\gamma + 1)\left((2\gamma + 1)^d\right)^{1/4}\sqrt{n}} =: 2s$$

  holds for $n \geq 1$, implying that $|F(\theta_1) - F(\theta_0)| \geq 2s > 0$.

We conclude the proof by Theorem B.5 using that $\alpha = \frac{5}{4}$ and $\max\left(\frac{e^{-\frac{5}{4}}}{4}, \frac{1-\sqrt{\frac{5}{8}}}{2}\right) = \frac{1-\sqrt{\frac{5}{8}}}{2}$.

---

[6]Notice that $(2\gamma + 1)^2 - \gamma^2 x > (2\gamma + 1)^2 - 4\gamma^2 x$, and $(2\gamma + 1)^2 - 4\gamma^2 x > 0 \Leftrightarrow x < \left(1 + \frac{1}{2\gamma}\right)^2$ for a positive $x$; hence the imposed assumption on $x$ ensures that the function $f_c$ is well-defined.

**Lower bound (ii): translation-invariant kernels.** Let $\Lambda_k$ denote the spectral measure associated to the kernel $k$ according to (1). Using the fact that $\mathrm{HSIC}_k(\mathbb{P}_{\theta_0}) = 0$, we have for $|F(\theta_1) - F(\theta_0)|$ that

$$
\left| F(\theta_1) - \underbrace{F(\theta_0)}_{=0} \right|^2 = F^2(\theta_1) = \mathrm{HSIC}_k^2(\mathbb{P}_{\theta_1}) = \mathrm{MMD}_k^2\left( \mathcal{N}(\boldsymbol{\mu}_1, \boldsymbol{\Sigma}_1), \mathcal{N}(\boldsymbol{\mu}_1, \boldsymbol{\Sigma}_0) \right)
$$

$$
\overset{(i)}{=} \left\| \psi_{\mathcal{N}(\boldsymbol{\mu}_1, \boldsymbol{\Sigma}_1)} - \psi_{\mathcal{N}(\boldsymbol{\mu}_1, \boldsymbol{\Sigma}_0)} \right\|_{L^2(\mathbb{R}^d, \Lambda_k)}^2
$$

$$
\overset{(ii)}{=} \int_{\mathbb{R}^d} \left| e^{i\langle \boldsymbol{\mu}_1, \boldsymbol{\omega}\rangle - \frac{1}{2}\langle \boldsymbol{\omega}, \boldsymbol{\Sigma}_1 \boldsymbol{\omega}\rangle} - e^{i\langle \boldsymbol{\mu}_1, \boldsymbol{\omega}\rangle - \frac{1}{2}\langle \boldsymbol{\omega}, \boldsymbol{\Sigma}_0 \boldsymbol{\omega}\rangle} \right|^2 \mathrm{d}\Lambda_k(\boldsymbol{\omega})
$$

$$
= \int_{\mathbb{R}^d} \underbrace{\left| e^{i\langle \boldsymbol{\mu}_1, \boldsymbol{\omega}\rangle} \right|^2}_{=1} \left| e^{-\frac{1}{2}\langle \boldsymbol{\omega}, \boldsymbol{\Sigma}_1 \boldsymbol{\omega}\rangle} - e^{-\frac{1}{2}\langle \boldsymbol{\omega}, \boldsymbol{\Sigma}_0 \boldsymbol{\omega}\rangle} \right|^2 \mathrm{d}\Lambda_k(\boldsymbol{\omega})
$$

$$
\overset{(iii)}{\geq} \int_A \left| e^{-\frac{1}{2}\langle \boldsymbol{\omega}, \boldsymbol{\Sigma}_1 \boldsymbol{\omega}\rangle} - e^{-\frac{1}{2}\langle \boldsymbol{\omega}, \boldsymbol{\Sigma}_0 \boldsymbol{\omega}\rangle} \right|^2 \mathrm{d}\Lambda_k(\boldsymbol{\omega}) \overset{(iv)}{\geq} \rho_n^2 \underbrace{\int_A \left[ h_{\boldsymbol{\omega}}'(0) \right]^2 \mathrm{d}\Lambda_k(\boldsymbol{\omega})}_{=:(2c)^2} \overset{(v)}{=} \underbrace{\frac{(2c)^2}{n}}_{=:(2s)^2 > 0} ,
$$

where $(i)$ holds by Sriperumbudur et al. [2010, Corollary 4(i)] (recalled in Theorem B.2). $(ii)$ follows from the analytical form $\psi_{\mathcal{N}(\boldsymbol{\mu}, \boldsymbol{\Sigma})}(\mathbf{t}) = e^{i\langle \boldsymbol{\mu}, \mathbf{t}\rangle - \frac{1}{2}\langle \mathbf{t}, \boldsymbol{\Sigma} \mathbf{t}\rangle}$ of the characteristic function of a multivariate normal distribution $\mathcal{N}(\boldsymbol{\mu}, \boldsymbol{\Sigma})$. For $(iii)$, we define the non-empty open set

$$
A = \left\{ \boldsymbol{\omega} = (\omega_1, \ldots, \omega_d)^\mathsf{T} \in \mathbb{R}^d : \omega_{d_1} \omega_{d_1+1} < 0 \right\} \subset \mathbb{R}^d,
$$

and use that the integration of a non-negative function over a subset yields a lower bound. In $(iv)$, fix $\boldsymbol{\omega} \in A$ and let

$$
h_{\boldsymbol{\omega}} : \rho \in [0, 1] \mapsto e^{-\frac{1}{2}\langle \boldsymbol{\omega}, \boldsymbol{\Sigma}(d_1, d_1+1, \rho) \boldsymbol{\omega}\rangle} \in (0, 1].
$$

Note that $h_{\boldsymbol{\omega}}(\rho) = e^{-\frac{1}{2}\left( \boldsymbol{\omega}^\mathsf{T} \boldsymbol{\omega} + 2\rho \omega_{d_1} \omega_{d_1+1} \right)}$; $h_{\boldsymbol{\omega}}$ is continuous on $[0, 1]$ and differentiable on $(0, 1)$. Hence for any $\rho \in (0, 1)$, by the mean value theorem, there exists $\tilde{\rho} \in (0, 1)$ such that

$$
h_{\boldsymbol{\omega}}(\rho) - h_{\boldsymbol{\omega}}(0) = \rho h_{\boldsymbol{\omega}}'(\tilde{\rho}) \geq \rho \min_{c \in [0, 1]} h_{\boldsymbol{\omega}}'(c).
$$

We have the first and second derivatives

$$
h_{\boldsymbol{\omega}}'(c) = -\omega_{d_1} \omega_{d_1+1} e^{-\frac{1}{2}\left( \boldsymbol{\omega}^\mathsf{T} \boldsymbol{\omega} + 2c\omega_{d_1}\omega_{d_1+1} \right)}, \quad h_{\boldsymbol{\omega}}''(c) = \omega_{d_1}^2 \omega_{d_1+1}^2 e^{-\frac{1}{2}\left( \boldsymbol{\omega}^\mathsf{T} \boldsymbol{\omega} + 2c\omega_{d_1}\omega_{d_1+1} \right)} > 0,
$$

which implies that $c \mapsto h_{\boldsymbol{\omega}}'(c)$ is a strictly increasing function of $c$ and that it attains its minimum at $c = 0$, that is,

$$
h_{\boldsymbol{\omega}}(\rho) - h_{\boldsymbol{\omega}}(0) \geq \rho h_{\boldsymbol{\omega}}'(0) > 0,
$$

where the 2nd inequality holds by $\rho > 0$ and $\boldsymbol{\omega} \in A$. This shows that

$$
\left[ h_{\boldsymbol{\omega}}(\rho) - h_{\boldsymbol{\omega}}(0) \right]^2 \geq \left[ \rho h_{\boldsymbol{\omega}}'(0) \right]^2,
$$

and the monotonicity of integration gives $(iv)$. For $(v)$, we note that the kernel $k = \otimes_{m=1}^M k_m$ is characteristic [Szabó and Sriperumbudur, 2018, Theorem 4] (recalled in Theorem B.4) as the $(k_m)_{m=1}^M$-s are characteristic. Thus, $\mathrm{supp}(\Lambda_k) = \mathbb{R}^d$ (see Sriperumbudur et al. [2010, Theorem 9]; recalled in Theorem B.3), implying that $\Lambda_k(A) > 0$. $(v)$ follows from the positivity of $h_{\boldsymbol{\omega}}'(0)$ (for any $\boldsymbol{\omega} \in A$), from the fact that the integral of a positive function on a set with positive measure is positive, and from our choice of $\rho_n = n^{-1/2}$.

Now, by taking the positive square root, we have

$$
|F(\theta_1) - F(\theta_0)| \geq \frac{2c}{\sqrt{n}} =: 2s. \tag{11}
$$

We conclude by the application of Theorem B.5 using that $\alpha = \frac{5}{4}$ and $\max\left( \frac{e^{-\frac{5}{4}}}{4}, \frac{1 - \sqrt{\frac{5}{8}}}{2} \right) = \frac{1 - \sqrt{\frac{5}{8}}}{2}$.

## 4.3 Proof of Corollary 1

We use the same argument as in the beginning of the proof of Theorem 1 in Section 4.2 but adjust the setting in which we apply Theorem B.5. Specifically, we now let $\Theta = \{\theta_a := C_{X_a} \, : \, X_a \sim \mathbb{P}_a, \, a \in \mathcal{A}\}$ with $C_X$ defined as in (2) be the set of covariance operators, use the metric $(x, y) \mapsto \|x - y\|_{\mathcal{H}_k}$ for $x, y \in \mathcal{H}_k$, and keep the remaining part of the setup the same. Hence, it remains to lower bound $\|C_{X_{\theta_1}} - C_{X_{\theta_0}}\|_{\mathcal{H}_k}$. By using that HSIC is the RKHS norm of the cross-covariance operator, we obtain that

$$\left\|C_{X_{\theta_1}} - C_{X_{\theta_0}}\right\|_{\mathcal{H}_k} \overset{(i)}{\geq} \Big| \underbrace{\left\|C_{X_{\theta_1}}\right\|_{\mathcal{H}_k}}_{=\mathrm{HSIC}_k\left(\mathbb{P}_{\theta_1}\right)} - \underbrace{\left\|C_{X_{\theta_0}}\right\|_{\mathcal{H}_k}}_{=\mathrm{HSIC}_k\left(\mathbb{P}_{\theta_0}\right)} \Big| = |F(\theta_1) - F(\theta_0)| \overset{(ii)}{\geq} 2s = \frac{2c}{\sqrt{n}},$$

where $(i)$ holds by the reverse triangle inequality, $F$ is defined as in Section 4.2, and $(ii)$ is guaranteed by (11) for $c > 0$. We conclude as in the proof of Theorem 1(ii) to obtain the stated result.

## Acknowledgments and Disclosure of Funding

This work was supported by the German Research Foundation (DFG) Research Training Group GRK 2153: Energy Status Data — Informatics Methods for its Collection, Analysis and Exploitation, and by the pilot program Core-Informatics of the Helmholtz Association (HGF).

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

## A Auxiliary Result

In this section, we collect an auxiliary result. Lemma A.1 presents an upper bound on the Kullback-Leibler divergence between multivariate normal distributions.

**Lemma A.1** (Upper bound on KL divergence). *Let $d = \sum_{m=1}^{M} d_m$, with $d_m \in \mathbb{N}_{>0}$ ($m \in [M]$). Fix $i \in [d_1]$. Let $j = i + 1$, $\mathbb{P}_{\theta_0} = \mathcal{N}(\mathbf{0}_d, \mathbf{I}_d)$, and $\mathbb{P}_{\theta_1} = \mathcal{N}(\boldsymbol{\mu}_1, \boldsymbol{\Sigma}_1)$, with $\boldsymbol{\mu}_1 = \frac{1}{\sqrt{dn}} \mathbf{1}_d \in \mathbb{R}^d$, and $\boldsymbol{\Sigma}_1 = \boldsymbol{\Sigma}(i, j, \rho_n) \in \mathbb{R}^{d \times d}$ defined as in (5) ($\rho_n \in (0,1)$). Then, for $2 \le n \in \mathbb{N}$,*

$$\mathrm{KL}(\mathbb{P}_{\theta_1}^n || \mathbb{P}_{\theta_0}^n) \le \frac{1}{2n} + \frac{n}{2} \frac{\rho_n^2}{1 - \rho_n^2}.$$

*In particular, for $\rho_n^2 = 1/n$, it holds that $\mathrm{KL}(\mathbb{P}_{\theta_1}^n || \mathbb{P}_{\theta_0}^n) \le \frac{5}{4}$.*

*Proof.* With $\boldsymbol{\mu}_0 = \mathbf{0}_d$ and $\boldsymbol{\Sigma}_0 = \mathbf{I}_d$, we obtain that

$$
\begin{aligned}
\mathrm{KL}(\mathbb{P}_{\theta_1}^n || \mathbb{P}_{\theta_0}^n) &\overset{(a)}{=} \sum_{i \in [n]} \mathrm{KL}(\mathbb{P}_{\theta_1} || \mathbb{P}_{\theta_0}) \\
&\overset{(b)}{=} \frac{n}{2} \left[ \mathrm{tr}(\boldsymbol{\Sigma}_0^{-1} \boldsymbol{\Sigma}_1) + (\boldsymbol{\mu}_0 - \boldsymbol{\mu}_1)^{\mathsf{T}} \boldsymbol{\Sigma}_0^{-1} (\boldsymbol{\mu}_0 - \boldsymbol{\mu}_1) - d + \ln\left( \frac{|\boldsymbol{\Sigma}_0|}{|\boldsymbol{\Sigma}_1|} \right) \right] \\
&= \frac{n}{2} \left[ \underbrace{\mathrm{tr}(\boldsymbol{\Sigma}_1)}_{=d} + \underbrace{\|\boldsymbol{\mu}_1\|_{\mathbb{R}^d}^2}_{=\frac{1}{n^2}} - d + \ln\left( \frac{1}{\underbrace{|\boldsymbol{\Sigma}_1|}_{\overset{(c)}{=} \frac{1}{1-\rho_n^2}}} \right) \right] \\
&= \frac{1}{2n} + \frac{n}{2} \ln\left( \frac{1}{1 - \rho_n^2} \right) \overset{(d)}{\le} \frac{1}{2n} + \frac{n}{2} \frac{\rho_n^2}{1 - \rho_n^2} \overset{(e)}{\le} \frac{5}{4},
\end{aligned}
$$

where (a) is implied by Lemma B.1, (b) follows from Lemma B.2, (c) follows from the definition of the determinant, (d) is the consequence of the inequality $\ln(x) \le x - 1$ holding for $x > 0$, and (e) holds for $n \ge 2$ and $\rho_n^2 = 1/n$ as

$$\frac{n}{2} \underbrace{\frac{1/n}{1 - 1/n}}_{\frac{1}{n-1}} \le 1 \iff \frac{n}{2} \frac{1}{n-1} \le 1 \iff n \le 2(n-1) \iff n \ge 2,$$

and in this case (for $n \ge 2$) one has that $\frac{1}{2n} \le \frac{1}{4}$. $\qquad \square$

## B External Theorems

For self-completeness, we include the external statements that we use. The well-known result by Bochner, stated in Theorem B.1, completely characterizes continuous bounded translation-invariant kernels. Theorem B.2 allows expressing MMD with continuous bounded translation-invariant kernels in terms of characteristic functions, and Theorem B.3 gives an equivalent condition for a continuous bounded translation-invariant kernel to be characteristic. Theorem B.4 connects characteristic kernels to characteristic product kernels and to $\mathcal{I}$-characteristic product kernels on $\mathbb{R}^d$ (we include only the part relevant to our paper for brevity). We recall Le Cam's method in Theorem B.5 and collect results on the Kullback-Leibler divergence in Lemma B.1 and Lemma B.2.

**Theorem B.1** (Bochner; Theorem 6.6; Wendland [2005]). *A continuous function $\kappa : \mathbb{R}^d \to \mathbb{R}$ is positive definite if and only if it is the Fourier transform of a finite nonnegative Borel measure $\Lambda$ on $\mathbb{R}^d$, that is,*

$$\kappa(\mathbf{x}) = \int_{\mathbb{R}^d} e^{-i \langle \mathbf{x}, \boldsymbol{\omega} \rangle} \, \mathrm{d}\Lambda(\boldsymbol{\omega}), \quad \text{for all } \mathbf{x} \in \mathbb{R}^d.$$

**Theorem B.2** (Corollary 4(i); Sriperumbudur et al. [2010]). *Let $k : \mathbb{R}^d \times \mathbb{R}^d \to \mathbb{R}$ be a continuous bounded translation-invariant kernel. Then, for any $\mathbb{P}, \mathbb{Q} \in \mathcal{M}_1^+(\mathbb{R}^d)$,*

$$\mathrm{MMD}_k^2(\mathbb{P}, \mathbb{Q}) = \|\psi_{\mathbb{P}} - \psi_{\mathbb{Q}}\|_{L^2(\mathbb{R}^d, \Lambda_k)}^2,$$

*with $\psi_{\mathbb{P}}$ and $\psi_{\mathbb{Q}}$ being the characteristic functions of $\mathbb{P}$ and $\mathbb{Q}$, respectively, and $\Lambda_k$ defined in (1).*

**Theorem B.3** (Theorem 9; Sriperumbudur et al. [2010]). *Suppose $k : \mathbb{R}^d \times \mathbb{R}^d \to \mathbb{R}$ is a continuous bounded translation-invariant kernel. Then $k$ is characteristic if and only if $\operatorname{supp}(\Lambda_k) = \mathbb{R}^d$, with $\Lambda_k$ defined as in* (1).

**Theorem B.4** (Theorem 4; Szabó and Sriperumbudur [2018]). *Suppose $k_m : \mathbb{R}^{d_m} \times \mathbb{R}^{d_m} \to \mathbb{R}$ is continuous bounded and translation-invariant kernel for all $m \in [M]$. Then the following statements are equivalent:*

*(i) $(k_m)_{m=1}^M$-s are characteristic;*

*(ii) $\otimes_{m=1}^M k_m$ is characteristic;*

*(iii) $\otimes_{m=1}^M k_m$ is $\mathcal{I}$-characteristic.*

The next statement follows directly from Tsybakov [2009, Eq. (2.9)] and Tsybakov [2009, Theorem 2.2].

**Theorem B.5** (Theorem 2.2; Tsybakov [2009]). *Let $\mathcal{X}$ be a measurable space, $(\Theta, d)$ is a semi-metric space, and $\mathcal{P}_\Theta = \{\mathbb{P}_\theta : \theta \in \Theta\}$ is a class of probability measures on $\mathcal{X}$ indexed by $\Theta$. We observe data $D \sim \mathbb{P}_\theta \in \mathcal{P}_\Theta$ with some unknown parameter $\theta$. The goal is to estimate $\theta$. Let $\hat{\theta} = \hat{\theta}(D)$ be an estimator of $\theta$ based on $D$. Assume that there exist $\theta_0, \theta_1 \in \Theta$ such that $d(\theta_0, \theta_1) \geq 2s > 0$ and $\operatorname{KL}(\mathbb{P}_{\theta_1} || \mathbb{P}_{\theta_0}) \leq \alpha < \infty$ for $\alpha > 0$. Then*

$$\inf_{\hat{\theta}} \sup_{\theta \in \Theta} \mathbb{P}_\theta \left( d\left(\hat{\theta}, \theta\right) \geq s \right) \geq \max\left( \frac{e^{-\alpha}}{4}, \frac{1 - \sqrt{\alpha/2}}{2} \right).$$

We have the following property of the Kullback-Leibler divergence for product measures [Tsybakov, 2009, p. 85].

**Lemma B.1** (KL divergence of product measures). *Let $\mathbb{P} = \otimes_{i=1}^n \mathbb{P}_i$ and $\mathbb{Q} = \otimes_{i=1}^n \mathbb{Q}_i$. Then*

$$\operatorname{KL}(\mathbb{P} || \mathbb{Q}) = \sum_{i \in [n]} \operatorname{KL}(\mathbb{P}_i || \mathbb{Q}_i).$$

The following lemma [Duchi, 2007, p. 13] shows that the Kullback-Leibler divergence of multivariate Gaussians can be computed in closed form.

**Lemma B.2** (KL divergence of Gaussians). *The KL divergence of two normal distributions $\mathcal{N}(\boldsymbol{\mu}_1, \boldsymbol{\Sigma}_1)$ and $\mathcal{N}(\boldsymbol{\mu}_0, \boldsymbol{\Sigma}_0)$ on $\mathbb{R}^d$ is*

$$\operatorname{KL}(\mathcal{N}(\boldsymbol{\mu}_1, \boldsymbol{\Sigma}_1) || \mathcal{N}(\boldsymbol{\mu}_0, \boldsymbol{\Sigma}_0)) = \frac{\operatorname{tr}(\boldsymbol{\Sigma}_0^{-1} \boldsymbol{\Sigma}_1) + (\boldsymbol{\mu}_0 - \boldsymbol{\mu}_1)^\mathsf{T} \boldsymbol{\Sigma}_0^{-1} (\boldsymbol{\mu}_0 - \boldsymbol{\mu}_1) - d + \ln\left( \frac{|\boldsymbol{\Sigma}_0|}{|\boldsymbol{\Sigma}_1|} \right)}{2}.$$

