# OpenReview forum: "The Minimax Rate of HSIC Estimation for Translation-Invariant Kernels"
_NeurIPS.cc/2024/Conference — NeurIPS 2024 poster_

### Official Review · Reviewer_g9Na · 2024-06-30

**Soundness:** 3
**Presentation:** 4
**Contribution:** 3
**Rating:** 7
**Confidence:** 4

**Summary:**

This paper studies the statistical property of Hilbert Schmidt Independence Criterion (HSIC). Specifically, under either Gaussian or continuous bounded translation-invariant characteristic kernel that is defined on $\mathbb{R}^d$, the paper prove that HSIC can be estimated at the optimal rate in the minimax sense. The minimax lower bound is obtained via the Le Cam's method, where the authors find two distributions that are close in the KL divergence sense, but is dissimilar in the HSIC sense. Their results further demonstrates that many empirical estimators in literature is minimax optimal.

**Strengths:**

This paper provides the first minimax optimal rate for HSIC learing, via obtaining the information theoretical lower bound. This is an important contribution the litearture since HSIC is widely used for independence test.

A minor contribution is that the authors derive the closed form formula of HSIC under the Gaussian setting.

In proving the lower bound for the translation invariant kernel case, the authors restrict the integration of to be computed in a subset of $\mathbb{R}^d$, thereby obtaining a lower bound on the HSIC between $P_{\theta_1}$ and $P_{\theta_0}$. To me this is novel.

**Weaknesses:**

While the authors explain how they obtain the lower bound via Le Cam's method, I would more appreciate if the authors can briefly explain the main technical challenge in proving the lower bound. It seems that the lower bound proof is a straight forward application of Le Cam's approach.

It seems that in Zhou et al., (2019), they also obtain the lower bound for the centred covariance operator $C_{XX}$. The definition of $C_{XX}$ is similar to the HSIC setting and the lower bound is almost the same. I would appreciate the authors to provide more discussion on the difference between the two settings and how the proof is different from Zhou et al., (2019).

**Questions:**

See weakness

---

> ### Author Rebuttal · Authors · 2024-08-05
>
> We thank the reviewer for the time and effort invested, and the kind review.
>
> Below we answer the questions in detail.
>
> - __Technical challenge.__ There are three main tools for deriving lower bounds in the minimax setting: Le Cam's method, Fano's method, and Assouad's lemma. The main technical challenge in applying these known tools is coming up with an adversarial distribution pair and then also showing that the assumptions of the respective statements are all satisfied. In our case, we chose Le Cam's method and showed that its assumptions are satisfied for the parameterized Gaussian distribution (5) and its instantiations in Line 239 in the case of HSIC.
> - __Difference to (Zhou et al., 2019).__ Indeed, the rate obtained by (Zhou et al., 2019) for the covariance operator is the same as the one that we obtained for HSIC. Intuitively, one would guess that estimating the covariance operator, an element in a tensor product RKHS, is more challenging than estimating HSIC, a real number. Surprisingly, this is not the case, as our results indicate. Moreover, by an application of the reverse triangle inequality, our result allows to recover their result (see Corollary 1 and its proof). Regarding differences in the proof: The distribution pair considered by (Zhou et al., 2019) has a product structure, which, similar to (Tolstikhin et al., 2016), does not allow to obtain a lower bound for HSIC estimation.
>
> We hope that this answers all the questions of the reviewer.
>
> __References.__
>
> Tolstikhin, I. O., Sriperumbudur, B. K., & Schölkopf, B. (2016). Minimax estimation of maximum mean discrepancy with radial kernels. Advances in Neural Information Processing Systems, 29.
>
> Zhou, Y., Chen, D. R., & Huang, W. (2019). A class of optimal estimators for the covariance operator in reproducing kernel Hilbert spaces. Journal of Multivariate Analysis, 169, 166-178.

---

> > ### Comment · Reviewer_g9Na · 2024-08-08
> >
> > Thanks for replying. I will maintain my score.

---

### Official Review · Reviewer_tUTX · 2024-07-10

**Soundness:** 3
**Presentation:** 3
**Contribution:** 3
**Rating:** 5
**Confidence:** 3

**Summary:**

In this work, the authors prove that the minimax optimal rate of HSIC estimation on $\mathbb R^{d}$ for Borel measures containing the Gaussians with continuous bounded translation-invariant characteristic kernels is $n^{-1/2}$.

**Strengths:**

Testing whether a pair of random variables are independent is the central problem in statistics or machine learning community.  There are lots of independence tests proposed in past years such as distance correlation, dynamic slicing, etc.

The author established the minimax lower bound \( \Omega(n^{-1/2}) \) of HSIC estimation with \( M \geq 2 \) components on \( \mathbb{R}^d \) with continuous bounded translation-invariant characteristic kernels. As this lower bound matches the known upper bounds of the existing "classical" U-statistic and V-statistic-based estimators, and that of the Nyström HSIC estimator, their result settles their minimax optimality.

Establishing minimax rates is often deemed a challenging task. Given that the result sounds solid, I consider this a noteworthy result.

**Weaknesses:**

Due to my ignorance, I may not be able to provide a sufficient evaluation of the importance of this problem. It would be easier for me to evaluate its importance if the author could offer more related literature and a comparison with them.

**Questions:**

There are some minor issues.

Can the author provide a more concrete definition of HSIC? For example, in equation (2), does it imply that we have fixed a decomposition
$d=d_{1}+...+d_{M}$ and $P_{m}$  is the marginal distribution of P on $R^{d_{m}}$.?
​

---

> ### Author Rebuttal · Authors · 2024-08-05
>
> We thank the reviewer for the time and effort invested, and the kind review.
>
> In the following, we answer the questions.
>
> - __Related work.__ The related work can be divided into lower bounding the rate of estimating (1) the kernel mean embedding, (2) maximum mean discrepancy, and (3) the covariance operator. The existing results do not permit establishing the minimax rate of HSIC estimation due to the following reasons.
> 1.  The estimation of the mean embedding (1) concerns the estimation of an element in an RKHS, which could be more difficult than the estimation of a real-valued function of it (MMD; Tolstikhin et al., 2016, or, in our case, HSIC).
> 2. The proof for obtaining the lower bound of (2) relies on a distribution pair in which both distributions factorize, that is, they are independent. The corresponding HSIC value is thus zero and Le Cam's method is not applicable; hence the existing proof does not address the setting of HSIC.
> 3. In the case of (3), one intuitively expects that estimating the real-valued HSIC is "easier" than estimating the covariance operator, which is an element in a tensor product RKHS. Our result sheds light on a surprising phenomenon: this intuition is false. Moreover, with Corollary 1, we are able to recover the lower bound on covariance operator estimation.
>
> These notes are also elaborated in Remark 1(c)--(e).
>
> - __Definition of HSIC.__ HSIC captures the dependencies of a probability measure by quantifying the discrepancy of the measure to the product of its marginals as the distance of the corresponding mean elements in an RKHS. This also corresponds to the norm of the cross-covariance operator, which (2) makes explicit; the equivalence holds for arbitrary kernel-enriched domains. In line with Line 121-122 and Theorem 1, and as correctly stated in the review, we consider $\mathcal{X} = \times_{m=1}^M \mathcal{X_m}$ with $\mathcal{X}=\mathbb{R}^d$ (for the domain of $\mathbb{P}$), $\mathcal{X}_m=\mathbb{R}^{d_m}$, i.e. one has the decomposition $d=d_1 + \cdots + d_M$, with $\mathbb P_m$-s being the corresponding marginals of $\mathbb{P}$ on $\mathbb R^{d_m}$ ($m=1,\ldots,M$). The associated kernels are $k_m:\mathbb R^{d_m} \times \mathbb R^{d_m} \to \mathbb{R}$.
>
> We hope that this answer clarifies all the questions of the reviewer.
>
> __References.__
>
> Tolstikhin, Ilya O., Bharath K. Sriperumbudur, and Bernhard Schölkopf. "Minimax estimation of maximum mean discrepancy with radial kernels." Advances in Neural Information Processing Systems 29 (2016).

---

> > ### Comment · Reviewer_tUTX · 2024-08-09
> >
> > Thanks for replying. I will maintain my score.

---

### Official Review · Reviewer_6YpR · 2024-07-15

**Soundness:** 3
**Presentation:** 2
**Contribution:** 3
**Rating:** 6
**Confidence:** 2

**Summary:**

The rate at which HSIC can be estimated is an important and open problem, in this paper, the authors prove that
the minimax optimal rate of HSIC estimation for Borel measures is $\mathcal{O}(n^{-0.5})$ with M>=2 components, which is very important as existing conclusion only holds for M=2. Other byproducts can be naturally introduced, implying the minimax lower bound for the estimation of
cross-covariance operator, which can be further specialized to get back the minimax result on the estimation of the covariance operator.

**Strengths:**

1. The paper answers an important while open problem which may be very important for the community and generalizes the existing result from M=2 to M>=2.
2. The paper's structure is clear and well organized.
3. The paper is solid and mathematically heavy, the proof is provided with details.

**Weaknesses:**

1. Overall, the paper is not easy to follow as the paper's main contribution seems to be the proof part.
2. I wouldn't say it is the weakness or the author's problem, as this is a theoretical paper, experiments are not necessary. Still is it possible to design toy experiments to validate the conclusions in the paper?

**Questions:**

NA

---

> ### Author Rebuttal · Authors · 2024-08-05
>
> We thank the reviewer for the time and effort invested, and for the kind review.
>
> To answer the question regarding experiments: In the minimax framework, one bounds the convergence rate from above and from below.
> - The former can be validated empirically in some cases. Indeed, for HSIC, one can compute the theoretical (= population) HSIC value for a fixed (kernel, distribution)-pair---for instance, for the Gaussian kernel with the Gaussian distribution (Lemma 1)---, and verify the known (Smola et al., 2007; Theorem 2) convergence rate of the estimator w.r.t. this value.
> - The latter, that is, the lower bound, can only be derived theoretically---as one must consider all possible estimators ($\inf_{\hat{F_n}}$ in (3)) and all possible probability distributions ($\sup_{P\in\mathcal{P}}$ in (3)) of the considered class ($\mathcal{P}$)---which is what we tackled in this article.
>
> Minor note regarding the point 'existing conclusion only holds for $M=2$': We are not aware of minimax guarantees for HSIC even for $M=2$; our minimax analysis handles both the case of $M=2$ and $M>2$ in a unified fashion (for $M \ge 2$).
>
> We hope that this settles the stated questions.
>
> __References.__
>
> Smola, A., Gretton, A., Song, L., & Schölkopf, B. (2007). A Hilbert space embedding for distributions. In International Conference on Algorithmic Learning Theory (pp. 13-31).

---

### Decision · Program_Chairs · 2024-09-25

**Decision:**

Accept (poster)

**Comment:**

The paper presents the minimax optimal rate of HSIC estimation on $R^d$ for Borel measures containing the Gaussian distributions with continuous bounded translation-invariant characteristic kernels. In particular, the result implies the optimality in the minimax sense of frequently used estimators, including the U-statistic, the V-statistic, and the Nyström-based method on $R^d$.

Reviewers all agree that the obtained result is novel, noteworthy, and of interest to the community. It is an interesting, though not particularly innovative methodologically, application of the well-known Le Cam's method for minimax risk.